# Influence of Sequential CAD/CAM Milling on the Fitting Accuracy of Titanium Three-Unit Fixed Dental Prostheses

**DOI:** 10.3390/ma14061401

**Published:** 2021-03-13

**Authors:** Doo-Bin Song, Man-So Han, Si-Chul Kim, Junyong Ahn, Yong-Woon Im, Hae-Hyoung Lee

**Affiliations:** 1Department of Biomaterials Science, College of Dentistry, Dankook University, 119 Dandaero, Cheonan 31116, Korea; yujong2804@daum.net (D.-B.S.); ajy3809@naver.com (J.A.); 2Department of Dental Laboratory Technology, Daejeon Health Institute of Technology, Daejeon 34504, Korea; mshan@hit.ac.kr; 3Department of Dental Laboratory Technology, Chungbuk Health & Science University, Cheongju 28150, Korea; iyoseb@chsu.ac.kr; 4Department of Dental Laboratory, Kyungdong University, Wonju 26495, Korea; iywoon9101@naver.com

**Keywords:** fitting accuracy, sequential milling, fixed dental prosthesis, bur wear, CAD/CAM

## Abstract

This study investigated the fitting accuracy of titanium alloy fixed dental prostheses (FDP) after sequential CAD/CAM (Computer Aided Design/Computer Aided Manufacturing) fabrication. A three-unit FDP model connecting mandibular second premolars and molars was prepared and scanned to fabricate titanium FDPs by CAD/CAM milling. A total of six FDPs were sequentially milled in one titanium alloy disk using a new set of burs every time (n = 4). The fitting accuracy of FDPs was mesiodistally evaluated by a silicone replica technique and the measurement was triplicated at four different locations: MO (marginal opening), MG (marginal gap), AG (axial gap), and OG (occlusal gap). Data were statistically analyzed using ANOVA and Tukey’s HSD test. The fitting accuracy of PMMA (polymethyl methacrylate) FDPs milled using the worn or new bur were evaluated by the same procedure (n = 6). The mean dimensions of titanium FDP for all measuring positions, except for AG, were significantly increased from the third milling. However, no difference was noted between the first FDP and the second FDP milled with the same set of burs. Severe edge chippings were observed in all milling burs. Detrimental effects of the worn burs on the fitting accuracy were demonstrated in the CAD/CAM-milled PMMA FDP. The results recommend proper changing frequency of cutting burs to achieve the quality of fit and predictable outcomes for dental CAD/CAM prostheses.

## 1. Introduction

Dental CAD/CAM (Computer Aided Design/Computer Aided Manufacturing) systems have become mainstream in the fabrication of most dental fixed prostheses, even for metal restorations that have conventionally been fabricated by casting with the lost-wax technique [1,2,3]. In addition to the milling technique, metal prostheses can currently be fabricated by selective laser melting using CAD data [4]. This translation was further encouraged by the following advantages of digitalized technologies: reduced production time and labor cost and less quality difference between dental laboratories [2,3,4,5].

The degree of marginal and internal accuracy of fixed dental prostheses (FDPs) is of great importance to evaluate the clinical performance of a newly developed fabrication system. Although the fit of metal restoration fabricated by conventional lost-wax casting is better than that of milled restoration, the marginal fit of titanium copings fabricated by the milling technique with CAD/CAM is currently comparable to that of the conventional casting technique with or without manual refinement [1,6]. Laser melting technology also exhibits a marginal and internal fit of metal crowns that is comparable to conventional production procedures, indicating a clinically acceptable fit [4]. However, inconsistencies in fitting accuracy have been noted for metal restorations fabricated by these two milling and/or sintering systems with conventional casting [7]. A direct comparison of these techniques showed a better fit for the laser melting technique compared with CAD/CAM or castings in a three-unit FDP model [5]. However, another study indicated that the laser melting technique resulted in relatively poorer accuracy than the milling or casting technique [8,9,10]. The CAD/CAM-milling technique produced Co-Cr copings ranging from 52 to 113 µm in the marginal opening regardless of the shape of the margin, which can be considered clinically acceptable at <120 µm [6,11].

Titanium and titanium alloys have many advantages as fixed and removable prosthetic materials in terms of physical properties and corrosion resistance [12]. However, these products are not disseminated clinically due to innate problems that mainly arise during conventional melt casting procedures. The difficulties in the fabrication of titanium dental prostheses could be solved with direct milling of Ti ingots by CAD/CAM technology. A precise internal fit is the most important requisite for successful FDP treatment. A previous study demonstrated that clinically acceptable marginal discrepancies for titanium single crowns could be obtained from various dental CAD/CAM systems [1]. However, the milling accuracy is strongly affected by the physical properties of CAD/CAM materials. In the machining of prosthetic materials with high hardness and toughness, such as base metal alloys, tool wear, or damage to the cutting edge of burs is common during continuous use, thwarting precise fitting of milled objects [13,14,15].

There have been concerns about the durability of milling tools (tungsten carbide burs) for dental CAD/CAM machines [16]. However, the effects of bur wearing on the fitting accuracy of CAD/CAM-fabricated metal prostheses have rarely been investigated. The aims of this study were to investigate the change in the marginal and internal fit of titanium three-unit FDP by sequential CAD/CAM fabrication. The null hypothesis was that no significant differences would be found in the accuracy of fit of CAD/CAM-fabricated FDP for continuous milling.

## 2. Experimental Procedures

A standardized three-unit posterior FDP model connecting mandibular second premolars and molars was manufactured with stainless steel cylinders. Two abutment teeth were circumferentially prepared with a 12° total occlusal convergence and 0.8-mm chamfer margin using a 6° tungsten carbide burs (Hopf, Ringleb & Co. GmbH & Cie., Berlin, Germany) and lab milling machine (F4, DeguDent GmbH, Hanau, Germany). A central fossa with a 1-mm depth was prepared in the center of the occlusal surface to simulate occlusal form of natural teeth, and the sharp edges were intentionally beveled to reduce possible needs for internal refinement [1]. All dimensions of the master model are shown in Figure 1A. The metal master mode was duplicated using silicon material (Dublisil 15, Dreve, Unna, Germany) and polyurethane die material (Polyurock, Metalor, Marin, Switzerland) to fabricate the working model. Six FDP working dies were individually digitalized using a lab scanner and converted into the respective FDP model (D700, 3Shape A/S, 3Shape, Copenhagen, Denmark) (Figure 1B). A cement space thickness of 30 µm was set with no space 1 mm from the margin. A standard three-unit FDP model conceptualized to have an occlusal surface was designed on each die using the 3 Shape Dental Designer program.

### 2.1. Titanium Three-Unit FDPs by Sequential Milling

The FDP CAD models were transferred to a 5-axis milling machine (DEG-5X100, Arum, Doowon ID, Daejeon, Korea) to fabricate the bridges using Titanium alloy CAD/CAM disk blocks (Ti-6Al-4V, *Φ*98 × 12 mm, Doowon ID). A total of six three-unit FDPs were sequentially milled in one Titanium disk using a new set of milling burs (Figure 1C). Using the individual CAD models, four Titanium disk blocks were milled with a new set of tungsten carbide ball-end milling burs (D 3.0, 2.0, 1.5, and 1.0 mm) every time (n = 4). After milling, the internal surfaces of the fabricated Ti FDPs were not subjected to any manual refinement to increase accuracy of fit.

### 2.2. Measurement of Fitting Accuracy

The fitting accuracy of the fabricated Ti FDPs on the two abutments was evaluated by a silicone replica technique using light-body and regular-body silicones [17]. The FDPs were seated with light-body polyvinyl siloxane (PVS, Smartsil, Seilglobal, Busan, Korea) on the corresponding dies under a constant loading of 50 N (A-001, MECC) to standardize the process (Figure 1D) [6]. After removal of FDP, the silicone film placed at the inner gap was fixed by pouring regular-body PVS (Charmflex, Denkist, Gunpo, Korea). The combined silicone replicas were cut mesiodistally to measure gap dimensions along with the deepest occlusal point. A wide variation in definition and terminology has been presented to evaluate the fitting accuracy of cast or CAD/CAM fixed prostheses [18,19,20,21]. Among those, this study measured the gap dimensions at four different locations: MO (marginal opening), MG (marginal gap), AG (axial gap), and OG (occlusal gap), which were applied from the study of Beuer et al. with a modification in the nomenclature (Figure 2) [19]. However, MG was measured vertically at the maximum concavity point of the inner margin for measuring consistency, which is approximately 0.4 mm distant from the MO location.

The silicon replicas were examined at 125× magnification using a metallurgical microscope (S39A, MIC, Saint Louis, MO, USA) and image analysis software (Topview, MIC, Saint Louis, MO, USA). The dimensions at each of four gap locations were determined by averaging data from the left and right sides of the premolar and second molar. All measurements were performed in triplicate with new silicon replicas to compensate for the limited number of specimens in this study [21]. To examine the effect of worn burs on milling accuracy, poly (methyl methacrylate) (PMMA) blanks (Vipi Block *φ*98.5 × 12 mm, Shin Dental Co., Seoul, Korea) for CAD/CAM fabrication were milled with the new set or the used set of burs for six Ti FDPs. The marginal opening and gap dimensions of the PMMA FDPs fabricated using the worn or new bur were evaluated by the same procedure (n = 6). Scanning electron microscopy (SEM, H-300, Hitachi, Tokyo, Japan) observations were conducted to examine the cutting surface of milling burs before and after use. The surface topographies of milled Ti FDPs were compared with SEM (Sigma 300, Zeiss, Jena, Germany) observations.

### 2.3. Statistical Analysis

Data were statistically analyzed using the IBM SPSS Statistics program (version 26, IBM, Armonk, NY, USA). The Levene test was used to test the homogeneity of variances between milling sequences for each measuring location. Mean data were compared by one-way ANOVA and a post hoc test (Tukey HSD) at the level of significance of *p* < 0.05.

## 3. Results

The marginal opening and gap dimensions at four measuring points, according to the milling sequence, are presented with a box-and-whisker plot in Figure 3. The mean MO dimensions of the first-milled and second-milled FDPs were 49 (±28) μm and 59 (±27) μm, respectively, demonstrating a high accuracy of fit that was comparable to those of a previous study [7]. However, the MO dimension considerably increased from the third milling FDP (172 ± 88 μm) to the sixth milling FDP (278 ± 90 μm), exceeding 120 μm, which is the clinically acceptable margin opening for a single crown [8,11,19,22]. The MG and OG milled by using the new bur set marked mean dimensions of 133 (±18) μm and 178 (±35) μm, respectively, which were considerably larger than their MO and AG values. Moreover, they showed a similar trend with the MO dimension, which was significantly increased from the third milling of Ti FDP. However, the mean gap dimension for AG (<40 μm) was unchanged throughout continuous milling for the six FDPs. Figure 4 presents the difference in marginal opening and gap of PMMA provisional FDPs milled between the as-received new burs and the worn burs used for six times milling of titanium FDP in the same CAD/CAM system, indicating a significantly decreased fit accuracy when PMMA FDPs are milled by using the damaged bur set (except for AG). After the sequential milling of six FDPs, severe damage appeared on the cutting edges of all diameter milling burs in scanning electron microscopy (SEM) observations (Figure 5). Figure 6 shows the boundary areas between the inner and outer milling surface at the margins of the first-milled and sixth-milled FDPs.

## 4. Discussion

The fitting accuracy of titanium three-unit fixed dental prostheses fabricated by a 5-axis dental CAD/CAM system was significantly affected by the milling sequence. The marginal opening and gap dimensions at all measuring positions, except for AG, severely deteriorate from the third milling with the same set of burs. No significant difference in the adaption accuracy of FDP was noted between the first and second millings. The overall results of this study rejected the null hypothesis, suggesting no significant differences in the fit accuracy of the three-unit FDP milled using the new or used burs. The declines in the internal fit accuracy of the titanium FDP during sequential milling could be attributed to damage or wear of the milling burs due to extended use over the limitation of the bur lifetime. The significant increase in the marginal opening and gap dimensions of the titanium FDP can be directly shown by the results of the PMMA three-unit temporary FDP between milled using as-received new burs and worn burs after the sixth milling of Ti prosthesis (Figure 4). The results of the present study are generally consistent with a recent study demonstrating a progressive increase in the marginal misfit of dental lithium disilicate ceramic crowns with the wearing diamond bur of the CAD/CAM unit, indicating that the reduced cutting efficiency significantly affects the fitting accuracy of milled prostheses [23].

The detailed reproduction of the crown intaglio surface by milling can be affected by the smallest bur diameter among the set of cutting tools, which must be considered in abutment preparation [15,24]. Furthermore, machining damage of cutting tools results in a reduction in tool diameter and a change in cutting-edge radius, as shown in Figure 5. Thus, machining hard materials, such as titanium and other dental base metal alloys, can easily cause bur wear and adaptation inaccuracy of milled prostheses [14,25]. A previous study demonstrated that, for fabrication of Co-Cr fixed prostheses, the milling method produced a relatively higher inaccurate internal fit than milled-wax casting or laser sintering techniques, indicating the influence of bur wearing [5]. Thus, the machinability of milled material can significantly influence the fitting accuracy of prostheses fabricated by dental CAD/CAM. However, the machinability is a multifactorial parameter [26,27].

The occlusal gap (OG) dimension was generally higher than those of the other measurement positions in most CAD/CAM-fabricated fixed dental prostheses as reported in this and other studies [5,6,19]. In this study, the reduced diameter of milling burs resulting from prolonged usage appeared to be more detrimental to the OG dimension than the other locations, whereas the AG dimension was not affected. An increased dimension of the OG can thicken the luting cement space, resulting in a weakening of the structural rigidity of dental fixed prostheses, especially for brittle materials [28]. Moreover, the worn milling burs with edge chippings produced a rougher surface and indistinct margin line in the titanium FDPs than did the new burs (Figure 6). The circumferential boundary areas (~300 um in width) that remained between the inner and outer milling surface at the margins of FDP are likely associated with different cutting directions during the CAM process, requiring future attention for marginal integrity of CAD/CAM-fabricated prostheses.

The results of this study recommend changing milling burs at a proper frequency to maintain the accuracy of fit for CAD/CAM fabrication of dental metal prostheses. Periodic changes in milling burs were required to achieve the quality of fit and predictable outcomes for dental CAD/CAM prostheses. Moreover, an improved durability of cutting edges in milling burs will reduce the fabrication costs for dental CAD/CAM prostheses with low machinable materials. A drill compensation program for bur wearing can be incorporated into dental CAD software to reduce the inaccuracy of dental prosthesis milling. The limitations of this study are that the fitting accuracy of sequential milling was evaluated with only one dental alloy and milling system. Further studies should evaluate the relation of the machinability of various dental alloys with the fitting accuracy for different dental CAD/CAM systems.

## 5. Conclusions

Within the limitations of this study, the marginal opening and gap dimensions of CAD/CAM-fabricated titanium three-unit FDP were significantly affected by the milling sequence or the conditions of milling burs. The results of this study recommend changing milling burs at a proper frequency to achieve the quality of fit and predictable outcomes for dental CAD/CAM prostheses.

## Figures and Tables

**Figure 1 materials-14-01401-f001:**
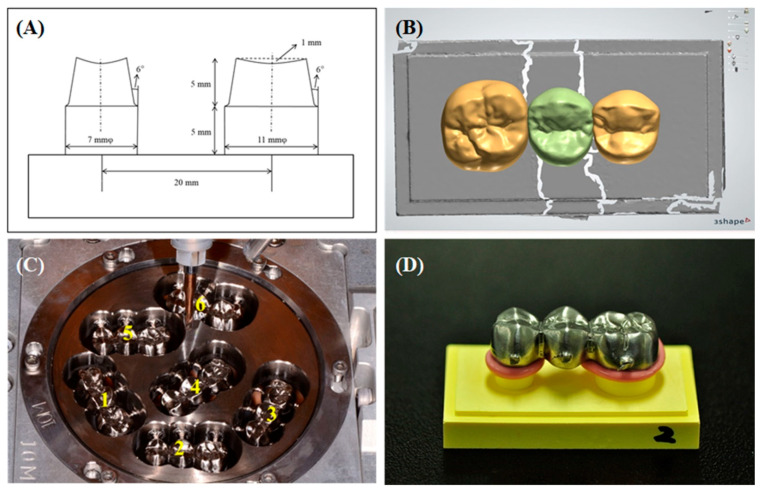
(**A**) Design of standard three-unit fixed dental prostheses (FDP), (**B**) designed FDP, and (**C**) sequentially milled Ti FDP on Computer Aided Design/Computer Aided Manufacturing (CAD/CAM). The numbers indicate the milling sequence. (**D**) Seated on FDP die with light-body silicone.

**Figure 2 materials-14-01401-f002:**
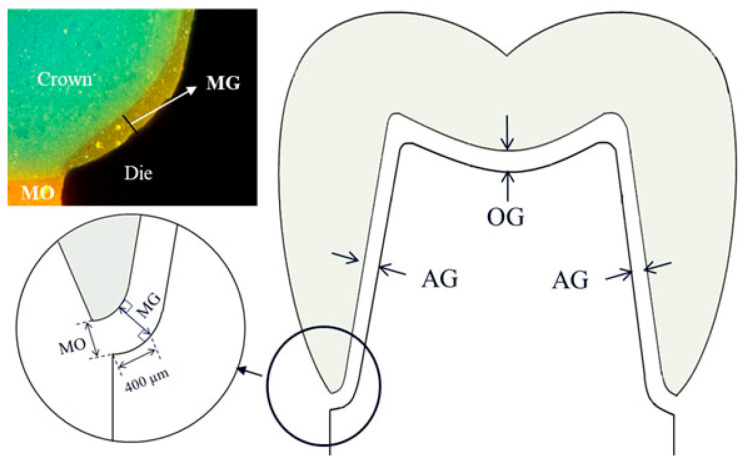
Diagram for measurement locations of MO (marginal opening), MG (marginal gap), AG (axial gap), OG (occlusal gap), and silicone replica.

**Figure 3 materials-14-01401-f003:**
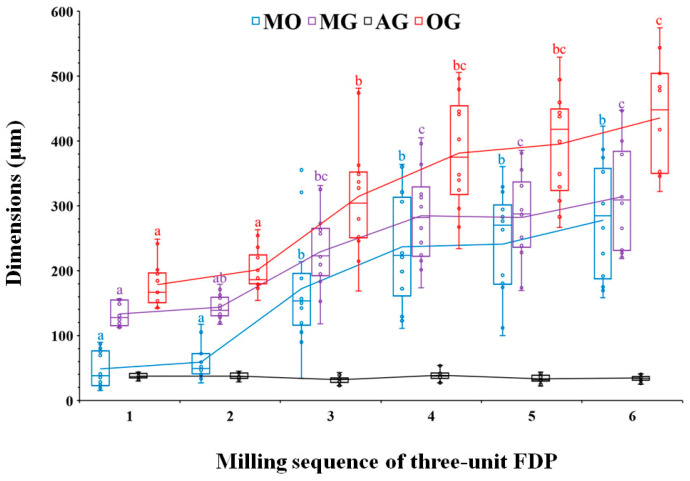
Box plot for marginal opening and gap dimensions of Ti-6Al-4V three-unit fixed dental prostheses (FDPs) according to the milling sequence. The same letters at each location indicate no statistical significance between the milling sequences at *p* < 0.05.

**Figure 4 materials-14-01401-f004:**
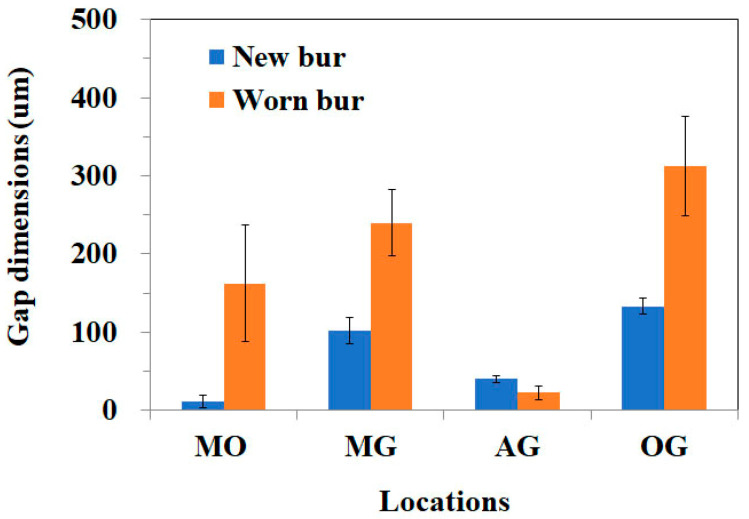
Difference in fitting accuracy of poly (methyl methacrylate) (PMMA) three-unit fixed dental prostheses (FDP) milled by as-received new bur and the worn burs used for milling of six titanium FDPs.

**Figure 5 materials-14-01401-f005:**
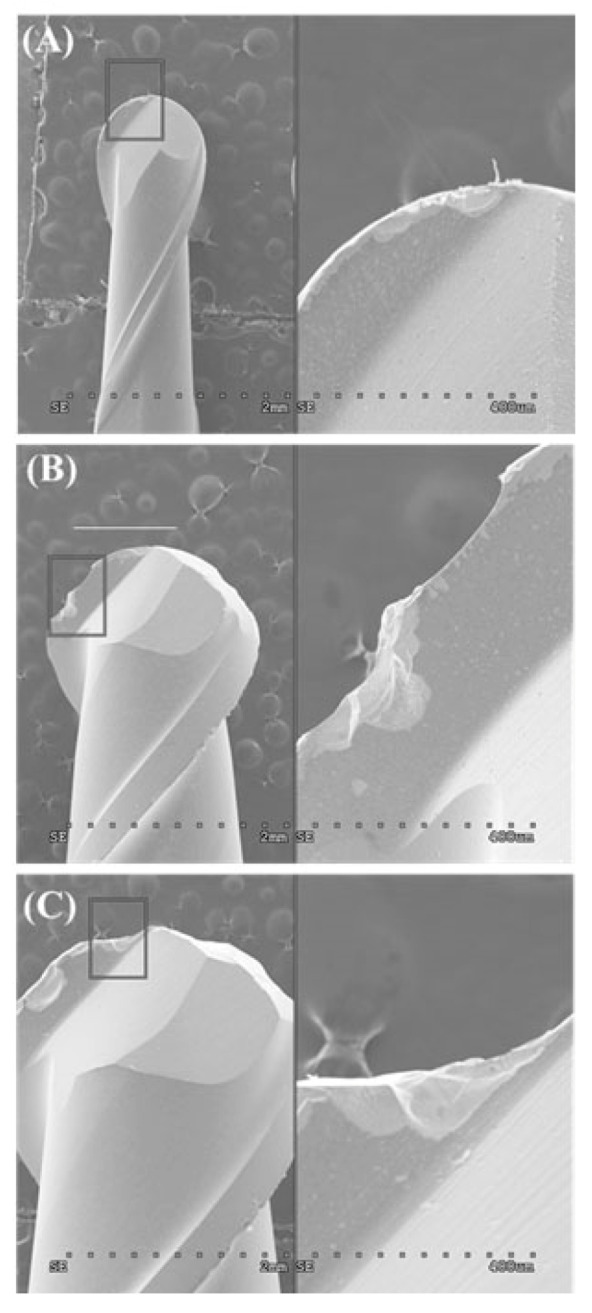
Edge chippings of milling burs with various diameters after milling of six three-unit FDPs. (**A**) 1.0 mm, (**B**) 2.0 mm, and (**C**) 3.0 mm.

**Figure 6 materials-14-01401-f006:**
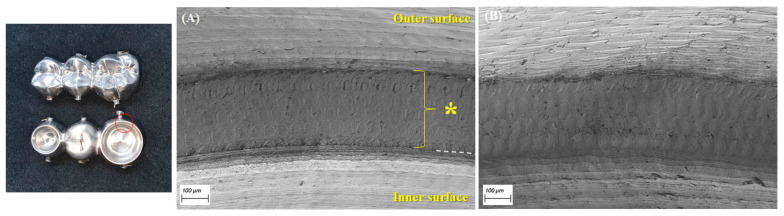
Scanning electron microscopy (SEM) photographs of boundary areas (asterisk marks) remained between the inner and outer milling surface at the crown margins (red circled area) of the first-milled FDP (**A**) and the sixth-milled FDP (**B**). The white dashed line (**A**) indicates the margin line at MO area.

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
