# Peer review of "Influence of Sequential CAD/CAM Milling on the Fitting Accuracy of Titanium Three-Unit Fixed Dental Prostheses"

_materials, 2021, doi:10.3390/ma14061401_

Round 1

Reviewer 1 Report

There are some weaknesses through the manuscript which need improvement. Therefore, the submitted manuscript cannot be accepted for publication in this form, but it has a chance of acceptance after a major revision. My comments and suggestions are as follows:

1- Abstract gives information on the main feature of the performed study, but some details about the experimental practice must be added.

2- Using a short title for the paper is suggested.

3- Authors must clarify necessity of the performed research. Aims and objectives of the study, and also differences with the previous researches must be clearly mentioned in the last part of introduction.

4- The literature study must be enriched. In this respect, authors must read and refer to the following papers: (a) https://doi.org/10.1016/j.jmbbm.2019.02.009 (b) https://doi.org/10.1016/j.polymer.2020.122485

5- Why this particular material was selected for the study? All figures must be illustrated in high quality.

6- As this study deals with experimental practice, it is necessary to present some real figures to show specimen under test conditions.

7- In its language layer, the manuscript should be considered for English language editing. There are sentences which have to be rewritten.

8- The conclusion must be more than just a summary of the manuscript. List of references must be updated based on the proposed papers. Please provide all changes by red color in the revised version.

Author Response

First of all, the authors would like to express their appreciation for the insight and valuable

comments that were provided, which helped improve the manuscript.

For Reviewer 1

1- Abstract gives information on the main feature of the performed study, but some details about the experimental practice must be added.

  • We added some details.

2- Using a short title for the paper is suggested.

  • Title of this paper was slightly modified for clear meaning, but because of complex dental nomenclature system, it seemed difficult to shorted the title.

3- Authors must clarify necessity of the performed research. Aims and objectives of the study, and also differences with the previous researches must be clearly mentioned in the last part of introduction.

  • We describe the purpose and differences of this study at the end of introduction.
  • There have been concerns about the durability of milling tools (tungsten carbide burs) for dental CAD/CAM machines [16]. However, the effects of bur wearing on the fitting accuracy of CAD/CAM-fabricated metal prostheses have rarely been investigated. The aims of this study were to investigate the change in the marginal and internal fit of titanium three-unit FDP by sequential CAD/CAM fabrication. The null hypothesis was that no significant differences would be found in the accuracy of fit of CAD/CAM-fabricated FDP for continuous milling.

4- The literature study must be enriched. In this respect, authors must read and refer to the following papers: (a) https://doi.org/10.1016/j.jmbbm.2019.02.009

(b) https://doi.org/10.1016/j.polymer.2020.122485

  • We selected important and valuable references for this study. Recommended references are different type of materials.

5- Why this particular material was selected for the study? All figures must be illustrated in high quality.

  • With advancement of dental CAD/CAM and implant, metallic prostheses such as titanium alloys became important again. (Line 62-73)

6- As this study deals with experimental practice, it is necessary to present some real figures to show specimen under test conditions.

  • Figure 1/5/6 are real photos under test conditions.

7- In its language layer, the manuscript should be considered for English language editing. There are sentences which have to be rewritten.

  • This manuscript was subjected to English editing service.

8- The conclusion must be more than just a summary of the manuscript. List of references must be updated based on the proposed papers. Please provide all changes by red color in the revised version.
à modified
Reviewer 2 Report

Dear authors:

The topic analyzed is very interesting. However, there are several criticisms regarding research problems, methodology, or discussion. Please see list below.

-Abbreviations should be defined in parentheses the first time they appear in the abstract, main text, and in figure or table captions and used consistently thereafter. See Instructions for authors.

-Please consult the ninth edition of The Glossary of Prosthodontic Terms (J Prosthet Dent 2017;117:e1-e105.) when revising your manuscript to make sure that the terminology that you use is current and correct (CAD/CAM should be CAD-CAM)

 Material and Methods:

-This section needs improvement and further detail.

-Please discuss how the authors determined that sample size was adequate. Was a power analysis performed?

-Did you digitize the same working model six times, or did you use a different working model in each digitization? How many working models have you used?

-Please introduce in the manuscript the citation of Figure 1B, 1C, 1D.

-It is not clear how many Ti FPDs you have milled. Please explain.

-Please clarify the following paragraph: “However, MO was measured vertically at the maximum concavity point of the inner margin for measuring consistency, which is approximately 0.4 mm distant from the MO location.”

-Did you take the measurements at the same point? How did you standardize the measurements? Please clarify.

-How many measurements were taken to assess the fit per specimen?

-It is not clear why you have milled PMMA. What was the objective? Please explain

-Please edit the following paragraph at the end of the Statistical Analysis: “Scanning electron microscopy (SEM; H-300, Hitachi) observations were conducted to examine the cutting….” It is part of the methodology.

Results

-Please change the location of the Figures 3, 4,5 and 6 to the Results section, and explain them appropriately

Discussion

-Introduce the limitations of the study in the Discussion section.

-You must provide a short paragraph specifying the individual contributions of the authors. See Instructions for authors.

-Please edit References. See Instructions for authors.

Author Response

For Reviewer 2

-Abbreviations should be defined in parentheses the first time they appear in the abstract, main text, and in figure or table captions and used consistently thereafter. See Instructions for authors.

à We made a correction through the manuscript. CAD/CAM (Computer Aided Design/Computer Aided Manufacturing)

-Please consult the ninth edition of The Glossary of Prosthodontic Terms (J Prosthet Dent 2017;117:e1-e105.) when revising your manuscript to make sure that the terminology that you use is current and correct (CAD/CAM should be CAD-CAM)

à For CAD/CAM or CAD-CAM, although the JPD glossary edition recommend to use CAD-CAM, but CAD/CAM is more prevailing in google survey. Moreover, reference paper of this journal (Marginal Accuracy and Internal Fit of 3-D Printing Laser-Sintered Co-Cr Alloy Copings. Materials (Basel). 2017;10.) and other numerous recent papers (Impact of Occlusal Intercuspal Angulation on the Quality of CAD/CAM Lithium Disilicate Crowns. Journal of Prosthodontics. 2020;29:219-25) also used the term “CAD/CAM”. Therefore, we sustained the term (CAD/CAM) in our manuscript for the consistency.

Material and Methods:

-This section needs improvement and further detail.

-Please discuss how the authors determined that sample size was adequate. Was a power analysis performed?

à In this study, six three-unit FDPs were sequentially milled from each of four Ti disk blocks. Thus, a total of twenty-four FDPS were milled and all measurements were performed in triplicate with new silicon replicas to compensate for the limited number of specimens in this study. According to ref. #[21], it was suggested that when investigating the marginal fit of fixed dental restorations, the smaller sample size can be compensated by a larger number of measurements per sample. Although we did not perform a power analysis, statistical difference of this study would be valid.

-Did you digitize the same working model six times, or did you use a different working model in each digitization? How many working models have you used?

à We prepared six FDP working dies (models) and they were individually digitalized using a lab scanner (D700, 3Shape A/S). (Line 86)

-Please introduce in the manuscript the citation of Figure 1B, 1C, 1D.

à all are added in the text

-It is not clear how many Ti FPDs you have milled. Please explain.

à In this study, six three-unit FDPs were sequentially milled from each of four Ti disk blocks. Thus, a total of twenty-four FDPS were milled and tested.

-Please clarify the following paragraph: “However, MO was measured vertically at the maximum concavity point of the inner margin for measuring consistency, which is approximately 0.4 mm distant from the MO location.”

à Thank you for this kind and valuable comments. In this part, MO has to be replaced with MG, which is approximately 0.4 mm distant from the MO location.

-Did you take the measurements at the same point? How did you standardize the measurements? Please clarify.

à The gap locations measured in this study have been referred in previous study of Beuer et a. as described in the manuscript, except of MG: Among those, this study measured the gap dimensions at four different locations: MO (marginal opening), MG (marginal gap), AG (axial gap), and OG (occlusal gap), which were applied from the study of Beuer et al. with a modification in the nomenclature (Figure 2) [19]. In their study, the locations explained as followings,

  1. Marginal opening (MO): The marginal opening at the point of closest approximation between the die and ceramic margin of the retainer.
  2. Chamfer area (CA): The internal adaptation of the retainer at the point of the biggest diameter.: we used MG instead of CA, with at the maximum concavity point of the inner margin for measuring consistency.
  3. Axial wall (AW): The internal adaptation of the crown walls at the midpoint of the axial wall (2 mm occlusal to the margin of the die).
  4. Occlusal adaptation (OA): The internal adaptation of the surface of the crown to the die at the midpoint from the facial and proximal.

-How many measurements were taken to assess the fit per specimen?

à At each point for specimens (n=4), 2 (left and right) x 2 (premolar and second molar) x 3 (triplicated measurement) = 12 measurements per MO, MG, and AG. OG = 6 measurements

-It is not clear why you have milled PMMA. What was the objective? Please explain

à To examine the effect of worn burs on milling accuracy, poly(methyl methacrylate) (PMMA) blanks (Vipi Block φ98.5 x 12 mm) for CAD/CAM fabrication were milled with the new set or the used (worn) set of burs after titanium milling. (Line 144~) The results can be a direct evidence that the worn bur deteriorates the marginal accuracy of prostheses, because PMMA can be easily milled without bur wear.

-Please edit the following paragraph at the end of the Statistical Analysis: “Scanning electron microscopy (SEM; H-300, Hitachi) observations were conducted to examine the cutting….” It is part of the methodology.

à moved

Results

-Please change the location of the Figures 3, 4,5 and 6 to the Results section, and explain them appropriately

à All Figures are appeared in the results section.

The marginal opening and gap dimensions at four measuring points according to the milling sequence are presented with a box-and-whisker plot in Figure 3. The mean MO dimensions of the first- and second-milled FDPs were 49 (±28) μm and 59 (±27) μm, re-spectively, demonstrating a high accuracy of fit that was comparable to those of a previ-ous study [7]. However, the MO dimension considerably increased from the 3rd milling FDP (172±88 μm) to the 6th milling FDP (278±90 μm), exceeding 120 μm, which is the so-called clinically acceptable margin opening for a single crown [8, 11, 19, 22]. The MG and OG milled by using the new bur set marked mean dimensions of 133 (±18) μm and 178 (±35) μm, respectively, which were considerably larger than their MO and AG values. Moreover, they showed a similar trend with the MO dimension, which was significantly increased from the third milling of Ti FDP. However, the mean gap dimension for AG (<40 μm) was unchanged throughout continuous milling for the six FDPs. Figure 4 presents the difference in marginal opening and gap of PMMA provisional FDPs milled by between the new burs and the worn burs in the same CAD/CAM system, indicating a significantly de-creased fit accuracy when PMMA FDPs are milled by using the damaged bur set (except for AG). After the sequential milling of six FDPs, severe damage appeared on the cutting edges of all diameter milling burs in SEM observations (Figure 5). Figure 6 shows the boundary areas between the inner and outer milling surface at the margins of the first- and sixth-milled FDPs.

Discussion

-Introduce the limitations of the study in the Discussion section.

à The limitations of this study are that the fitting accuracy of sequential milling was evaluated with only one dental alloy and milling system. (Line 258)

-You must provide a short paragraph specifying the individual contributions of the authors. See Instructions for authors.

à added

-Please edit References. See Instructions for authors.

à corrected

Round 2

Reviewer 1 Report

The paper has been improved and the current version can be considered for publication.

Reviewer 2 Report

Dear authors,

Thank you very much for your responses. Certainly, the paper has been improved.

Although your say in your letter that certain items have been corrected, remain the same in the text.

Perhaps, you have another version of the manuscript.

Please see list below.

Results

-Please change the location of the Figures 3, 4, 5 and 6 to the Results section, and explain them appropriately.

Please insert your figures in the main text after the paragraph of its first citation, following the Instructions for authors.

Discussion

-Please edit the references number in the Discussion section.

-"It is not clear why you have milled PMMA. What was the objective? Please explain".

The authors have answered the question. Please introduce your explanation in the text.

References

-Please edit References. Use the Font type and size indicated in the Materials-template (Palatino Linotype, 9).